# Really Doing Great at Estimating CATE? A Critical Look at ML Benchmarking Practices in Treatment Effect Estimation

**Alicia Curth**
University of Cambridge
amc253@cam.ac.uk

**David Svensson & James Weatherall**
AstraZeneca

**Mihaela van der Schaar**
University of Cambridge
UCLA
The Alan Turing Institute

## Abstract

The machine learning (ML) toolbox for estimation of heterogeneous treatment effects from observational data is expanding rapidly, yet many of its algorithms have been evaluated only on a very limited set of semi-synthetic benchmark datasets. In this paper, we investigate current benchmarking practices for ML-based conditional average treatment effect (CATE) estimators, with special focus on empirical evaluation based on the popular semi-synthetic IHDP benchmark. We identify problems with current practice and highlight that semi-synthetic benchmark datasets, which (unlike real-world benchmarks used elsewhere in ML) do not necessarily reflect properties of real data, can systematically favor some algorithms over others – a fact that is rarely acknowledged but of immense relevance for interpretation of empirical results. Further, we argue that current evaluation metrics evaluate performance only for a small subset of possible use cases of CATE estimators, and discuss alternative metrics relevant for applications in personalized medicine. Additionally, we discuss alternatives for current benchmark datasets, and implications of our findings for benchmarking in CATE estimation.

## 1 Introduction

Modeling treatment effect heterogeneity has become a hot topic in machine learning (ML) research over the last five years [1–10]. This is not surprising: most of empirical science ultimately aims to assess the effect of treatments, policies or other interventions, and interest is slowly moving from the population average to the individual in many applications. A particularly prominent motivating example (which motivates also the authors of this paper) is personalized medicine and the goal to develop treatment strategies for each individual patient [11]. In this context, ML-algorithms which estimate conditional average treatment effects (CATE) have many more use cases than 'just' providing estimates of (personalized) effects: they can be used to provide decision support (e.g. providing treatment recommendations[12]), to find the right patients for a treatment (e.g. subgroup selection in clinical trials[13]) or for scientific discovery (e.g. discovering effect modifiers [14]).

Yet, while the majority of ML research in CATE estimation has been on deep models [1, 2, 5–10], applications in substantive fields have continued to rely on tree-based methods, e.g. causal forests [15, 16] in econometrics [17, 18] or model-based recursive partitioning [19] and the forest-based virtual twins approach [20] in statistics in medicine. Even though some of this can surely be attributed to a time lag in adoption of novel methodology and reluctance to change, we believe that hesitation in adoption may also be rooted in insufficient and unconvincing empirical evaluation of such algorithms thus far, both in terms of datasets and metrics that were considered.

35th Conference on Neural Information Processing Systems (NeurIPS 2021) Track on Datasets and Benchmarks.

When new algorithms are proposed within the ML community, it is common practice to empirically evaluate their performance against existing *baseline algorithms* on *benchmark datasets*, which, in other areas of ML, often consist of a body of real data e.g. ImageNet [21]. When trying to evaluate estimators of CATE on real data, the fundamental problem is that – even if strict identifying assumptions hold – ground truth individual treatment effects are never observed [22]. That is because, in reality, we can only ever *either* observe an individuals 'potential' outcome under treatment – $Y(1)$ – *or* without treatment – $Y(0)$–, but never both; yet the estimation target is the expected value of the (unobserved) difference $Y(1) - Y(0)$. Thus, to provide proof-of-concept, papers proposing new CATE estimators have largely relied on synthetic or semi-synthetic datasets (with real covariates), in which both potential outcomes are simulated and hence known, to showcase their properties. As is common in other areas of ML [23], a few of these (semi-synthetic) datasets have emerged as de facto 'standard' benchmarks which have been used in a wide range of ML papers in the last years. The most prominent example of this is unquestionably [24]'s Infant Health Development Program (IHDP) benchmark; since its first use in [1, 2], it has been used in almost all algorithmic papers on binary heterogeneous treatment effect estimation published at major ML conferences (e.g. [3–10, 25–27]).

As we discuss below, this common practice of blindly relying on standard benchmarking tasks has recently encountered criticism in other areas of ML (e.g. [23, 28, 29]). In this paper, we argue that current ML benchmarking practices in CATE estimation not only suffer from some of the same problems, but have a crucial additional problematic dimension that the literature seems to be oblivious to: semi-synthetic benchmarks like IHDP are treated the same as real-world benchmarks prevalent elsewhere in ML, yet, because the data are partially simulated, such benchmarks necessarily encode some assumptions on problem structure (which do not necessarily reflect characteristics that are likely to appear in real applications) in the underlying data generating process (DGP) and may *systematically favour some types of algorithms* over others.

**Outlook.** To verify this, we present an in depth case study of the IHDP benchmark (Section 3), and highlight that there is a range of problems with its current use. While some of these issues are unique to the IHDP benchmark, we present additional evidence that some issues – e.g. the systematic advantage of some algorithms over others due to how the data is generated – are prevalent in semi-synthetic benchmark datasets more generally by analyzing some of the ACIC2016 [30] simulations (Appendix D). Because a central goal of this paper is to raise awareness of the limitations inherent to the *current use* of generic *semi-synthetic* benchmark comparisons in the ML CATE literature, we then discuss (dis-)advantages of semi-synthetic datasets and alternative approaches with the aim to encourage more fair and insightful benchmarking practices and to provide a starting point for researchers looking to *use* or *develop* new benchmarks (Section 4). Further, we reconsider *what* is being evaluated, and discuss other metrics that might be more interesting for practical applications and how they lead to new requirements on benchmark datasets (Section 5). We close by discussing implications (Section 6).

*Remark:* The critiques presented here are specifically aimed at current evaluation practices in ML methods papers, i.e. papers proposing CATE estimation algorithms published at ML conferences. This excludes papers published in substantive fields and methods papers published in e.g. econometrics and statistics, which follow different standards for empirical evaluation (see Appendix B).

## 1.1 Related work

**Critiques of benchmarking practices in other ML sub-fields.** There is a broader movement across ML critizising existing benchmarking practices, e.g. for lack of documentation and understanding of contexts of dataset creation [31, 32], label inconsistencies [33], undervaluing data quality more generally [34], ignorance for socio-technical context of datasets[29] and the overall fragility of the benchmarking process [23]. A common point of criticism that is particularly applicable to our context is that (possibly flawed) datasets and associated performance metrics often unintentionally become *the* de facto benchmark task for a problem due to use in a seminal paper in the field because reviewers subsequently expect replication of results [23, 29]. We would argue that this is indeed exactly what happened with the IHDP benchmark: The seminal work of [1, 2] introduced the ML community to both the CATE estimation problem and the benchmark (which had previously been used only in [24] in statistics) unintentionally making it *the* ML benchmark for CATE estimation. Further, [23] also found the relative performance of algorithms on real-world benchmarks for natural language processing, computer vision, reinforcement learning and transformers to be sensitive to the considered task and

dataset. This is similar to our point that the DGPs in semi-synthetic benchmarks may inherently favour some algorithms, yet the key difference in our context is that such (dis-)advantages are systematic (partially foreseeable) and artificially created through the choice of DGP, instead of naturally inherent to a real application as in [23].

**Alternative evaluation frameworks proposed in the treatment effect literature.** Next to fully synthetic 'toy' DGPs designed for individual method papers (e.g. [9, 10, 15, 35]), a number of semi-synthetic benchmarks (based on real covariates but using simulated outcomes) have been created for the Atlantic Causal Inference Competitions (ACIC) [30, 36–38] and for CATE benchmarking papers in econometrics [39] and statistics in medicine [40]. Of these, only the ACIC2016 benchmark has been used repeatedly for evaluation of CATE estimators in the ML literature (e.g. in [9, 26, 41, 42]). Additionally, new *frameworks* for benchmarking of CATE estimators have been proposed recently; using generative models [43, 44], interventional datasets [45] or biased subsampling of RCTs [46]. By showcasing why current practice is problematic and potentially harmful, our paper is complementary to this line of work: we aim to argue that adoption of new benchmarking practices is indeed necessary. Therefore, we discuss the (dis)advantages of different proposals in more detail in section 4.

## 2 Background: Problem Setup and Learning Algorithms

We operate under the standard setup in the potential outcomes (PO) framework [47]. That is, we assume that any individual, associated with (pre-treatment) covariates $X \in \mathcal{X}$, has two potential outcomes $Y(0)$ and $Y(1)$ of which only $Y = Y(W) = WY(1) + (1 - W)Y(0)$, the outcome associated with the administered (binary) treatment $W \in \{0, 1\}$, is observed. We are interested in $\tau(x)$, the conditional average treatment effect (CATE), which is the expected difference between an individual's POs (conditional on covariates), i.e.

$$\tau(x) = \mathbb{E}[Y(1) - Y(0)|X = x] = \mu_1(x) - \mu_0(x) \tag{1}$$

where $\mu_w(x) = \mathbb{E}[Y(w)|X = x]$ is the expected PO. Identification of causal effects from observational data requires imposition of *untestable* assumptions; here, we rely on the strong ignorability conditions [48], which ensure that CATE is identifiable and can be estimated from observed data because $\mathbb{E}[Y(w)|X = x] = \mathbb{E}[Y|W = w, X = x]$ (i.e. – conditional on covariates – treated and untreated units are exchangeable). In Appendix A, we give an extended overview of the problem setup, necessary assumptions and unique problem characteristics.

**The problem with real data and why we need to simulate for benchmarking.** The fundamental problem of causal inference [22] – the absence of one of the POs in practice – makes benchmarking CATE estimators inherently more difficult than standard supervised learning algorithms: the ground truth label $Y(1) - Y(0)$ for any observation with $X = x$ is unknown in practice, making it impossible to test performance using real world benchmarks without further assumptions. Instead, to empirically evaluate estimators, the treatment effect estimation literature has historically relied on simulated outcome data, in which both POs are known and identifying assumptions hold because treatment assignment can be controlled. The most common approach for simulating outcomes seems to be to rely on data generating processes (DPGs) with additive effects (see e.g. the simulations used in [9, 10, 14–16, 20, 35, 49, 50]), i.e. to let $\mu_w(x) = f_0(x) + wf_\tau(x)$; that is, to assume that the treated regression surface $\mu_1(x)$ can be *additively decomposed* into a component shared with the control group ($f_0(x)$) and $f_\tau(x)$, an offset function determining treatment effect and heterogeneity. Additionally, $f_\tau(x)$ is often a simpler function than the treated surface $f_1(x)$ itself. Instead of additive transformations, it would also be possible that the PO regression surfaces have more general relationships, e.g. $\mu_1(x) = g(\mu_0(x))$ with $g(\cdot)$ some transformation function – in the IHDP benchmark, for example, $g(\cdot)$ is a logarithmic transformation – however, this specification seems less popular than additive parametrizations used in most DGPs in related work.

**Learning Algorithms and how DGPs can determine their performance.** A plethora of ML-based CATE estimators have been proposed in recent years. Here, we distinguish them along two key axes[1]: (i) the underlying *ML method* and (ii) the *estimation strategy*. The former is straightforward and refers simply to the ML method used to implement an algorithm, e.g. a random forest. The latter is

---

[1]Here, we focus on two axes most relevant to our case studies; an additional modeling dimension outside the scope of this paper would be the handling of confounding-induced covariate shift, e.g. by learning balancing representations [2] or propensity weighting [6, 9, 51]

crucial in the CATE context but has received relatively little explicit attention in the ML literature. As in [10, 52], we distinguish between two types of estimation strategy: *indirect* estimators that target the POs, i.e. first obtain estimates $\hat{\mu}_w(x)$ and then simply set $\hat{\tau}(x) = \hat{\mu}_1(x) - \hat{\mu}_0(x)$, and estimators that target CATE *directly*, e.g. by using pseudo-outcomes or other two-step procedures [10, 35, 49, 53].

From a theoretical viewpoint, we expect that the type of underlying DGP will (at least partially) determine which combination of ML method and estimation strategy will be most successful on any benchmark dataset. To see this, let $\epsilon_{sq}(\hat{f}(X)) = \mathbb{E}[(\hat{f}(X) - f(X))^2]$ denote the expected MSE for an estimate $\hat{f}(x)$ of a function $f(x)$ and consider the behaviour of $\epsilon_{sq}(\hat{\tau}(X))$ for different strategies. For indirect estimators, we have that $\epsilon_{sq}(\hat{\tau}(X)) \leq 2(\epsilon_{sq}(\hat{f}_1(X)) + \epsilon_{sq}(\hat{f}_0(X)))$, so that the error rate on the more complex of the regression surfaces will determine performance [4, 10]. Some direct estimators, on the other hand, can instead reach the same performance as a supervised learning algorithm with (hypothetical) target $Y(1) - Y(0)$, so that they can attain the error rate associated with $f_\tau(x)$ (see [10, 49, 53]). Whenever $f_\tau(x)$ is easier to estimate than the more complex of $f_1(x)$ and $f_0(x)$ (e.g. due to being sparser or smoother [10, 53]), direct learners thus have a clear theoretical advantage – which diminishes as $f_1(x)$ and $f_0(x)$ become less similar (making $f_\tau(x)$ increasingly complex). Further, the properties of the underlying ML method used to implement any estimation strategy will determine how well different types of functions $f_w(x)$ and/or $f_\tau(x)$ can be fit using a finite sample of observed data from a specific DGP.

## 3 A Close Look at Current Practice: A Case Study of IHDP

**Motivating example.** A popular baseline algorithm for CATE estimation from the statistics community is Wager & Athey (2018)'s [15, 16] Causal Forest (CF), which – unlike most ML estimators – targets CATE directly, comes with a set of theoretical guarantees and has already been applied in real empirical studies [17, 18]. Nonetheless, the seminal work of Shalit et al. (2017) [2] (and many extensions, e.g. [9, 51, 54]) shows that the neural network (NN)-based TARNet and its many variants outperform CF by lengths on the IHDP benchmark: across 1000 replications, [2] report an average RMSE of $0.88(0.02)$ for TARNet but $3.8(0.2)$ for CF. We asked ourselves a simple question: *Why?* Is it because TARNet is a uniformly better estimator? Or is it because TARNet is a NN while CF is a random forest (RF)? Is it maybe because TARNet models the potential outcomes, while CF models CATE directly? Is it the used metric? Or is it something else inherent to DGP or implementation?

**IHDP benchmarking practice.** The IHDP benchmark simulates outcomes based on *real* covariates ($n = 747, d = 25$) and treatments of the Infant Health and Development Program, a randomized experiment targeting an intervention which provides specialist child care to premature infants with low birth weight, originally discovering a positive effect on cognitive test scores [55]. To mimic an observational study, [24] introduced confounding and imbalance ($n_0 = 608, n_1 = 139$) by excluding a non-random proportion of treated individuals (those with nonwhite mothers), leading to incomplete overlap for the control group. The popular benchmark uses a DGP labelled setup 'B' in [24], with

$$\mu_0(x) = \exp((x + A)\beta) \text{ and } \mu_1(x) = x\beta - \omega \qquad (2)$$

where $\beta$ is a sparse coefficient vector with entries sampled from $(0, 0.1, 0.2, 0.3, 0.4)$ with probabilities $(0.6, 0.1, 0.1, 0.1, 0.1)$, $A$ is a fixed offset matrix, and $\omega$ is set uniquely in every simulation run, ensuring that the average treatment effect on the treated (ATT), which was the main estimand of interest in [24], is equal to 4. [2] created benchmarks based on 100 and 1000 realizations of the DGP, with pre-determined train-test splits (90/10). To evaluate the performance in estimating CATE, both [24] and [2] report averages of PEHE ('Precision in Estimating Heterogeneous Effects') – the RMSE of estimating $\tau(x)$, i.e. $\sqrt{n_{test}^{-1} \sum_{i=1}^{n_{test}} (\hat{\tau}(x_i) - \tau(x_i))^2}$, – across realizations.

**Experimental setup.**[2] We present a case study comparing the empirical performance of estimators relying on (i) different ML-methods (NNs and RFs[3]) and (ii) different estimation strategies (direct and indirect) to examine the sources of performance differences between the RF-based, direct estimator

---

[2]Code to replicate all experiments is available at `https://github.com/AliciaCurth/CATENets`

[3]In principle, many other base-methods could have been considered for this study; other popular CATE estimators rely on BART [24, 56], Gaussian Processes [3, 4] and GANs [5]. Here, we focus on RFs due to their popularity in applied research and NNs due to their popularity in the recent ML CATE literature.

Causal forest (CF) [15, 16] and the NN-based, indirect estimator TARNet[4] [2]. We use [2]'s IHDP-100 dataset (100 realizations of the DGP) and report out-of-sample performance.

We consider two (model-agnostic) indirect estimation strategies which are commonly referred to as T- and S-learner [49]; the former fits **t**wo separate regression surfaces for each treatment arm, while the latter fits a **s**ingle model in which $W$ is included as a standard covariate. We consider standard RF-based implementations (**TRF** and **SRF**) and a NN-based T-learner (**TNet**). Instead of a standard NN-based S-learner we use **TARNet** (as it can be seen as a hybrid of S- and T-learner [10]). As direct estimators, we consider [16]'s **CF**, which relies on a two-stage procedure solving a local moment equation inspired by the Robinson transformation [57], and, as a NN-based variant, we use [35]'s R-learner (**RNet**), which relies on the same principle. For all forest-based methods we use the R-package `grf` [58] and for all NN-based methods we use the python-implementations `catenets` [10]. We use all models off-the-shelf; for all RFs this entails using 2000 trees, and all NNs have hyperparameter settings similar to those used in [2] for the IHDP experiments. We repeat each run 5 times with different seeds for all models. For further details, refer to the the Appendix.

### 3.1 Empirical findings

• **Preliminaries: Clearing up misconceptions about the IHDP benchmark.** We would like to begin by emphasizing that *IHDP is not 'one dataset'*: As different numbers of variables en-

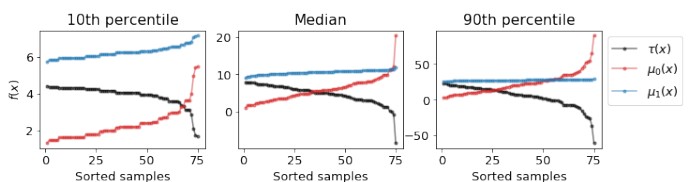

Figure 1: $\tau(x)$ and $\mu_w(x)$ in 3 test-set realizations with $\sigma_\tau$ at the 10th, 50th and 90th percentile across runs.

ter the response surfaces in each realization, there are systematic differences across simulation runs – a fact that is rarely taken into account. When the shared linear predictor is small – in runs in which $\beta$ is randomly more sparse – the exponential in $\mu_0(x)$ and linear function in $\mu_1(x)$ will run almost parallel, making $\tau(x)$ close to constant. Conversely, many nonzero entries in $\beta$ will lead to more exponential behaviour in $\mu_0(x)$ – making $\tau(x)$ highly nonlinear. While [24] fixed the magnitude of the ATT across runs through an offset, CATE is thus allowed to vary freely, leading to $\sigma_\tau = \sqrt{Var_{test}(\tau(x))}$ (capturing the spread of CATE within the test-set of a run) differing by orders of magnitude, ranging between $0.04$ and $51.51$, across the 100 simulations.

Fig. 1 shows $\tau(x)$ and $\mu_w(x)$ for runs with small/medium/large $\sigma_\tau$, indeed exhibiting the expected behaviour. We also note a further qualitative difference: in runs with small $\sigma_\tau$, *all* individuals benefit from treatment, yet for large $\sigma_\tau$, some individuals experience a very large *negative* effect. In the context of the intervention studied in the original IHDP experiment, it seems plausible that individuals with high $\mu_0(x)$ would have smaller $\tau(x)$ *in magnitude*, yet the *large negative* effects observed in runs with large $\sigma_\tau$ do not seem to be compatible with the original results of [55]. To the best of our knowledge, *the outcomes were indeed not created to mimic a specific realistic scenario*, rather setup 'B' (eq. 2) was only one of *multiple* DGPs used in [24] to compare different ATT estimators.

•**Finding (i): Reporting simple averages of RMSE across simulation runs appears inappropriate.** In Fig. 2 we plot a histogram of test RMSE across the 100 realizations of the simulation, in which it is obvious that the right tail of scores very long and heavy. Averaging over such scores gives extremely high weight to the few runs with very high RMSE, which, as we show below, are the runs with large $\sigma_\tau$. The common practice to simply report averaged RMSE across all runs gives algorithms that perform well mainly for the few runs with large $\sigma_\tau$ a clear advantage (which is problematic due to a

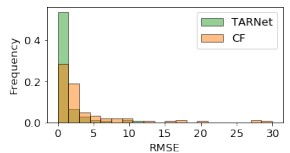

Figure 2: Histogram of test RMSE of CATE estimation for TARNet and CF on IHDP.

lack of justification why these runs *should* be more important, especially because – as outlined above – these runs appear to be the least realistic); thus it might be more appropriate to consider alternative metrics to aggregate scores across runs (e.g. normalized RMSE or paired sample test statistics).

---

[4]Here, we focus on [2]'s TARNet instead of CFRNet (which adds a balancing term to handle selection on observables) because the latter adds an *additional modeling dimension* on top of the two dimensions whose effect we wish to isolate in this paper. Further, the performance improvement of CFRNet over TARNet (0.71 vs 0.88) is marginal relative to the gap between CF and TARNet discussed above.

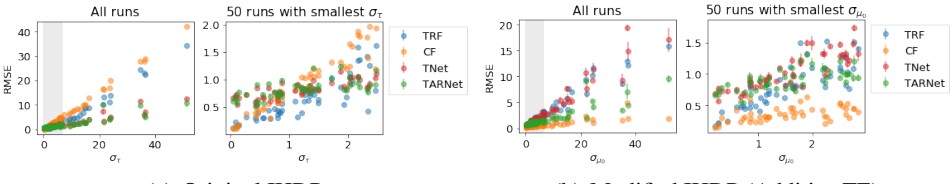

(a) Original IHDP                    (b) Modified IHDP (Additive TE)

Figure 3: Out-of-sample RMSE of CATE estimation across 100 IHDP draws (original and modified setting) for TNet, TARNet, TRF and CF. Averaged across 5 runs, bar indicates one standard error. Shaded area in left plots indicates area which are zoomed on in right plots.

•**Finding (ii): Comparing the estimators under consideration across all runs, indirect strategies appear to perform better than the direct alternative and NNs appear to outperform RFs.** In the left panel of Fig. 3(a), we plot RMSE by $\sigma_\tau$ for TARNet, TNet, TRF and CF on IHDP-100. We omit RNet and SRF for readability, refer to Appendix D.1 for full results. We observe that the considered direct estimators consistently perform worse than their indirect alternatives. This is not unexpected given the DGP and the theoretical arguments in the previous section: $\mu_0(x)$ and $\mu_1(x)$ are not similar (on an additive scale) in the underlying DGP, making $\tau(x)$ a difficult function to estimate directly. Thus, direct learners have no expected theoretical advantage in this setting; absent such advantage, we conjecture that the two-stage nature of such algorithms may hurt finite sample performance relative to indirect estimators as errors could accumulate across stages. Further, considering only the left panel of Fig. 3(a), it appears as if NN-based estimators have a clear advantage on this dataset and the underlying ML-method seems to matter much more than the CATE estimation strategy used. TARNet also consistently outperforms TNet; this is not surprising given that the linear predictor $X\beta$ is shared across both outcome surfaces, making for a perfect shared representation that TARNet can exploit.

•**Finding (iii): Relative performance systematically differs across runs.** Recall that simulation runs in which more covariates have (large) nonzero coefficients have higher CATE heterogeneity and magnitude. In the left panel of Fig. 3(a), it is obvious that as the measured variation in CATE increases, the absolute discrepancy between methods becomes more extreme. Due to high differences in magnitude, this perspective masks more interesting differences that become most apparent when considering *relative* differences using only the simulation runs with less extreme $\sigma_\tau$. In the right panel of Fig. 3(a) we observe that the considered forest-based estimators perform *better* than the NN-based versions for $\sigma_\tau$ small, and that the discrepancy between direct and indirect learners is less extreme.

•**Finding (iv): Tree-based methods suffer in the tails due to the exponential.** The findings above let us speculate that the poor performance of the forest-based estimators (which can be thought of as adaptive nearest neighbor methods [15])

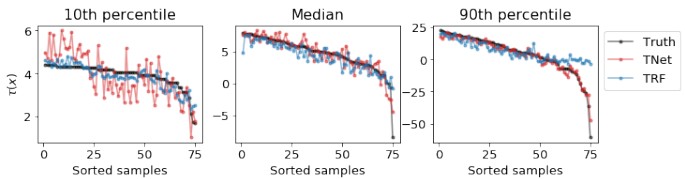

Figure 4: True and predicted CATE for TNet and TRF, on 3 datasets with $\sigma_\tau$ at the 10th, 50th and 90th percentile across runs.

may be partially caused by boundary bias on exponential specifications. We confirm this in Fig. 4 where we plot T-learner estimates of $\tau(x)$ for the 3 realizations shown in Fig. 1. We observe that for large $\sigma_\tau$, major performance discrepancies indeed arise only for the small subset of individuals with the largest linear predictor, which is where the exponential is the steepest and RFs hence cannot extrapolate.

•**Finding (vi): Tweaking the DGP slightly by creating an additive TE leads to completely different results.** Finally, to further test finding (ii), i.e. whether the observed performance differences across estimation strategies are indeed due to $\tau(x)$ *not* being simpler to estimate than the $\mu_w(x)$, we slightly alter the original simulation. We use $\mu_0^*(x) = \mu_0(x)$ and $\mu_1^*(x) = \mu_1(x) + \mu_0(x)$; i.e. the treatment effect is now additive and simple (linear). In Fig. 3(b) we report the RMSE of estimating CATE by $\sigma_{\mu_0}$ (as we found the variance induced by the exponential specification in $\mu_0(x)$, and not $\sigma_\tau$, to drive variation across runs). We observe that in this setting, almost all conclusions on relative performance are indeed *reversed* from what we observed in 3(a): CF performs best throughout, the considered direct learners perform better and NNs no longer have a clear advantage over RFs (possibly because the boundary bias now appears on *both* regression surfaces, which can difference out). This provides additional evidence that performance may indeed be partially determined by the fit of learning algorithm and the assumptions underlying the DGP.

## 3.2  Conclusion Case study

The empirical investigation above allows us to resolve our questions asked in the motivating example: The advantage of TARNet over CF on the IHDP benchmark has multiple sources; the DGP underlying the IHDP benchmark indeed appears to favour both the considered (i) NNs over RFs and (ii) indirect over direct estimators. Further, CF performs worse on runs with extreme $\sigma_\tau$, which effectively get much higher weight than the runs with very low $\sigma_\tau$ where it performs best. We conclude that the IHDP benchmark has multiple limitations, some of which are unique to its DGP and use – e.g. the heavy tail in the RMSE scores. Other problems are inherent to the use of semi-synthetic data more generally: in Appendix D.2, we perform a similar case study of a subset of the (semi-synthetic) ACIC2016 benchmark simulations and find that also here some algorithms are systematically favoured by characteristics that were embedded in the DGPs used for simulation. Additionally, we contrast the algorithms' performance on the two semi-synthetic datasets with their performance on the real-world Twins dataset [59] in Appendix D.3.

## 4  Creating alternative benchmarks

### 4.1  (Dis)Advantages of simulating fully synthetic outcomes

In the previous section, we presented some evidence that would seem to speak against the use of (semi-) synthetic datasets for benchmarking: we consider 'credibility' of such benchmarking results a major problem, as we showed that synthetic DGPs can inherently favour some class of algorithms over others. One could argue that any real world data can also inherently favour some algorithms [23]; however, the nature of favoured algorithms would usually not be known a priori, and any real data would at least reflect a scenario similar to what an algorithm will encounter when deployed – which is not necessarily the case for hand-crafted DGPs. Further, we have ourselves experienced in the past that coming up with a good set of synthetic response surfaces from scratch is extremely difficult: there are infinite possibilities for combining functional forms for outcome generation with CATE-specific experimental knobs, e.g. the degree of CATE heterogeneity and structure of confounding. Exhaustively considering all possible combinations would not only be computationally prohibitive, but might also not provide insightful experimental evidence for use in practice given that only a subset might realistically appear in the wild.

Nonetheless, we think that – if inherent limitations of (semi-)synthetic benchmarks are taken into account and properly discussed when interpreting results – datasets with simulated outcomes can be very useful for benchmarking, especially when the goal is to understand performance differences of algorithms: Simulations allow to directly study the effects of relevant experimental knobs, e.g. sample size, functional forms, confounding, and the degree and form of effect heterogeneity. Additionally, domain knowledge from target applications (e.g. on plausible functional forms, the degree of effect heterogeneity and form of confounding) could be used to select the most likely scenarios among a plethora of possible simulations. As we discuss below, it would also be possible to simulate response surfaces according to models fit to some real data. Further, high quality real data (which we discuss below) can be extremely difficult to come by and make publicly available, making simulations a good solution for reproducible, open science. Finally, the 'data-blind' benchmarking culture in ML can also lead to misinterpretation of findings based on real-world data [29], so that simulations can be a safer and more ethical choice if the goal is only to showcase performance differences.

### 4.2  Alternatives to simulating fully synthetic response surfaces

Below, we discuss four alternative benchmarking approaches used or proposed in recent literature. All have advantages and disadvantages, leading us to conclude that *no* approach appears bulletproof.

• **1.  Simulate only the treatment effect and use real baseline outcomes.**  In CATE estimation, only change in outcome due to treatment, and not the baseline outcome itself, is of key interest. To construct datasets with baseline outcomes of realistic complexity, it is therefore possible to use untreated (continuous) outcomes from a real dataset (or outcomes from datasets without treatments) as $Y(0)$ and add only a simulated $\tau(x)$ to obtain $Y(1) = Y(0) + \tau(x)$, resulting in known CATE $\tau(x)$. Such partially simulated POs (using simulated treatment assignments) were used in [39].
**Advantages:** This approach preserves some realism in outcomes, works with any real data even if no treatments are available, and allows the investigator to systematically vary only the target function

(instead of the nuisance functions $\mu_w(x)$).

**Disadvantages:** Similar to fully synthetic outcome functions, this approach can lack credibility as the simulated $\tau(x)$ may be unrealistic and could inherently favour some algorithms by design.

**• 2. Use models fit on real data to generate outcomes and act as ground truth.** Instead of hand-crafting possibly unrealistic response surfaces, one could fit any PO model on factual (real) data, and then generate new, ground truth POs for all observations in the sample using the fitted model (e.g. [40, 60]). [43] and [44] take this approach one step further, and suggest fitting generative models not only for the POs, but also for the covariates, allowing to generate completely new samples.

**Advantages:** This approach could be interpreted as calibration of synthetic DGPs to real data; it effectively corresponds to choosing the best simulation model within the considered outcome model class (e.g. RFs or NNs). As suggested by [44], additional explicit experimental knobs can be incorporated through post-fitting manipulation of learned outcome or treatment mechanisms.

**Disadvantages:** This approach may also lack credibility because resulting benchmarks can be expected to favour algorithms that are most closely related to those that generated the data.

**• 3. Use biased subsampling of RCT data to construct observational studies.** To assess performance of algorithms under (observed) confounding in observational settings, [46] propose to create benchmarks by performing biased subsampling of real RCT datasets.

**Advantages:** This approach retains real outcomes, and can be used to evaluate how well confounding is tackled in ATE and PO estimation.

**Disadvantages:** Even in RCTs, $Y(1) - Y(0)$ is unobserved, thus this approach lacks a mechanism to directly evaluate CATE estimates (unless $X$ is discrete and low-dimensional, so that there can be multiple observations for each $X = x$). Nonetheless, as treatment is randomized, the Horvitz-Thompson transformation [61] could be used as an unbiased (yet noisy) proxy for evaluation as in e.g. [62].

**• 4. Use 'all potential outcomes' data.** In some situations, it is possible to construct (proxies of) counterfactuals from real data, making a real contrast $Y(1) - Y(0)$ available for evaluation of estimators (while only one outcome is used at training time). The Twins benchmark dataset used in [5, 59], in which Twins represent counterfactuals, is the only such benchmark in current use that we are aware of (see Appendix D.3 for a description of the dataset and some experimental results). [45] highlight other applications in which multiple potential outcomes can be derived for essentially identical units, including gene regulatory networks and software systems. Additionally, we would argue that in some cases, pre- and post-treatment observations from longitudinal datasets could serve as proxies for counterfactuals[5], e.g. if treatments act quickly and observations are recorded at sufficiently small time-intervals (so that the necessary assumption 'nothing changed except for treatment'[22] is credible), or if data from cross-over trials [63] is available.

**Advantages:** Such approaches essentially allow to overcome the fundamental problem of absence of $Y(1) - Y(0)$ while preserving credibility of the benchmarking results as all used data is real.

**Disadvantages:** In many applications, such data may be hard to come by. Each dataset also represents only *one* specific real world scenario which will be domain-specific and not necessarily generalizable; additionally, no experimental knobs are available.

## 5   Considering Alternative Evaluation Metrics

So far, we have discussed only *the datasets* which algorithms are evaluated on, and not *what* is evaluated. Next, we move to argue that current benchmarking practice has also led to a focus on a restricted set of evaluation metrics, which evaluate performance only for a subset of possible use cases of CATE estimators. Most papers report only the PEHE, which corresponds to evaluating only the *statistical estimation quality* of the CATE function itself[6]. This would be most relevant for applications where the exact value of individual CATE estimates are used directly, e.g. for statistical reporting. In many practical applications, however, it is not the exact CATE estimate that is of ultimate interest, but quantities and decisions derived from it [11, 12]. Below, we give two examples of possible downstream tasks and implied performance metrics with practical relevance in the context of drug development and personalized medicine.

---

[5]Note that we suggest these approaches only to obtain realistic proxies of counterfactuals for benchmarking purposes in methods research, not as an identification strategy in substantive applications.

[6]In a similar vein, the absolute bias in ATE estimation is sometimes also reported due to its use in [2]

• **Do estimators correctly identify drivers of effect heterogeneity?** In medicine, one distinguishes patient characteristics by whether they have *prognostic* value – providing information on outcomes *regardless of treatment status* – or *predictive* value – providing information on the *differential effect of treatment* [13, 64]. Being able to correctly recognize the two types of covariates is key for clinical and research settings: predictive information may be interesting from a purely scientific perspective, can be used in drug development, for definition of subgroups and for assignment of treatments in practice, while prognostic information may e.g. be used in trial planning [14, 50, 65]. It can therefore be of great interest to assess whether CATE estimators correctly identify predictive information, i.e. drivers of effect heterogeneity, instead of relying on spurious correlations. This can be evaluated e.g. by using variable importance measures to assess the proportion of correctly top-ranked predictive covariates [14, 66]. Using this approach, [14] found that CFs and xgboost-based T-learners often mistake prognostic variables for predictive ones, and that good performance in terms of RMSE of CATE does not necessarily translate to good performance in terms of identifying predictive information.

• **Do estimators lead to correct treatment rules?** In the presence of predictive biomarkers with *qualitative* interactions, there exist patients who benefit and patients who are harmed by treatment [13]; if the ultimate goal is to use a CATE estimator for *decision support* it can thus be relevant to test whether a model correctly 'classifies' such patients [12]. Possible metrics include the value $E[Y(d(X))] = E[d(X)Y(1) + (1 - d(X))Y(0)]$ [67] of the implied treatment rule $d(X) = \mathbb{1}\{\hat{\tau}(x) > \delta\}$ for threshold $\delta$ ([2] report the related *policy risk* only for the real-world Jobs dataset) or evaluation of rankings implied by $\hat{\tau}(x)$ using e.g. the area under the receiver-operating curve with classification target $\mathbb{1}\{\tau(x) > 0\}$ [68].

Relative to commonly considered 'experimental knobs' (usually the degree of confounding, as in e.g. [2, 9]) – the metrics outlined above highlight that there are relevant additional dimensions to DGPs that should be varied in benchmark experiments. Evaluating the discovery of drivers of heterogeneity, for example, requires ground truth knowledge of variable status and varying degrees of predictive and prognostic information, which can e.g. be achieved by conducting a range of simulations as in [14]. Current benchmarking practice, however, does not allow to assess performance on this task, and may even inadvertently discourage research in this direction. This is because popular 'black-box' benchmark datasets that were created through repeated random sampling of DGPs (IHDP, ACIC2016) do not record which variables enter each response surface, so that no ground truth is available for performance assessment. Further, due to the special DGP used in IHDP, there is no predictive/prognostic distinction – all important variables are *both* predictive and prognostic – so that models explicitly searching for such structure will underperform on this benchmark (see [10] for an example of this), making this an unattractive modeling strategy and research direction if evaluation on standard benchmarks is required. Other metrics also reveal important DGP dimensions; e.g. to evaluate treatment rules implied by a CATE estimator, it would be important to *systematically* vary the amount of harm versus benefit a treatment brings across a population, instead of having it vary uncontrolled as a by-product of other changes in the DGP.

*Remark: Oracle vs factual metrics.* PEHE and alternatives above are 'oracle metrics' as they cannot be computed by practitioners during deployment (they require an unknown *ground truth*). It could thus be insightful to, where possible, report 'factual metrics', e.g. performance on predicting factual observations or other recently proposed factual validation criteria (e.g. [25, 41]), alongside 'oracle metrics': if relative factual performance correlates well with oracle metrics, this can provide heuristics for model selection in practice; if not, this can make practitioners aware of possible fallacies.

## 6 Conclusions and takeaways

A central goal of this paper was to raise awareness of the limitations inherent to the *current use* of generic semi-synthetic benchmark comparisons in the ML CATE literature. Our case studies highlighted that for semi-synthetic benchmark datasets – e.g. IHDP – in which (a component of) the DGP is known, some estimators can have an *expected* advantage over others, due to a better fit of underlying ML method and/or estimation strategy with the assumptions underlying the generation of the POs. We do not consider this finding surprising, yet it is rarely taken into account (or is at least not explicitly discussed) when benchmark datasets and baseline algorithms are selected for use in related work. 'Data-blindness', especially w.r.t. how data was generated, can thus be harmful and misleading; it can result in ignorance for systematic disadvantages some types of algorithms may have on a benchmark (which, as in the case of IHDP, does not even necessarily reflect characteristics of

real data). This is particularly problematic in a community like ML, where a lack of 'state-of-the-art' results on popular benchmarks can prohibit publication and follow-up research, leading to inadvertent 'overfitting' of research to a specific benchmark [23]. Below, we discuss additional takeaways.

• **No type of dataset is without flaws, thus acknowledging limitations is essential.** We do not aim to discourage the use of (semi-)synthetic data in general: on the contrary, we have argued that using simulated outcomes can have many advantages. Instead, we conclude that *all* data sources have inherent downsides, and would argue that *increased awareness and acknowledgement of possible limitations* of used datasets (by both authors *and* reviewers) is most crucial to ensure *fair benchmarking* of CATE estimators and *correct interpretation* of empirical results, e.g. by discussing whether any included algorithms are known to be inherently (dis)advantaged on a particular dataset. Conducting experiments across multiple types of benchmarks (e.g. considering both *a range* of synthetic setups and a real 'all potential outcomes' dataset) may lead to most convincing conclusions.

• **Interaction with domain experts is instrumental for credible benchmarking.** We consider increased collaboration with domain experts from relevant applications essential for creation of any type of new CATE benchmark dataset and task, either (i) to curate and make publicly available real data, (ii) to understand which simulation settings are more (less) realistic in different domains or (iii) to understand the relevance of different metrics of success. Further, domain knowledge will be crucial to move from ideal conditions to new tasks and corresponding datasets incorporating additional complications prevalent in real data – e.g. missing data [69], noise in (subjective) outcomes [70], limited compliance [71] and other types of endpoints (e.g. time-to-event outcomes [72]) – which are not considered in any current ML benchmark dataset for CATE estimation.

• **Insightful benchmark comparisons require careful choice of baseline algorithms.** Our case studies have shown that sensible baseline algorithms are necessary to understand drivers of performance. Because many DGPs will favour one class of algorithms over another, we consider it important for authors to provide *principled* instead of 'apples to oranges' baseline comparisons – the latter may mask the sources of performance gain. That is, we consider some baseline comparisons more insightful than others. Instead of the common practice of direct comparisons of estimators that vary along *multiple dimensions*, e.g. TARNet and CF in our motivating IHDP example, we would, for example, be interested whether a newly proposed learning strategy outperforms existing learning strategies implemented using *the same underlying ML method*, or conversely, whether a new ML method applied to the CATE estimation context outperforms commonly applied ML methods using *the same (pre-existing) learning strategy*.

## Acknowledgments and Disclosure of Funding

Some parts of this paper were previously presented in "Doing Great at Estimating CATE? On the Neglected Assumptions in Benchmark Comparisons of Treatment Effect Estimators" (Curth & van der Schaar, 2021) [73] at the workshop on Neglected Assumptions in Causal Inference at ICML 2021. We thank workshop attendants, anonymous reviewers (at both the workshop and NeurIPS21) as well as members of the vanderschaar-lab for many insightful comment and suggestions. AC gratefully acknowledges funding from AstraZeneca. Additionally, MvdS received funding from the Office of Naval Research (ONR) and the National Science Foundation (NSF, grant number 1722516).

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
