# Really Doing Great at Estimating CATE? A Critical Look at ML Benchmarking Practices in Treatment Effect Estimation
# Appendix

**Alicia Curth**
University of Cambridge
amc253@cam.ac.uk

**David Svensson & James Weatherall**
AstraZeneca

**Mihaela van der Schaar**
University of Cambridge
UCLA
The Alan Turing Institute

This appendix is organized as follows: We first present additional background on heterogeneous treatment effect estimation, including an additional overview of considered methods (Section A) and then briefly discuss how CATE benchmarking practices in ML differ from other fields (Section B).We then provide further experimental details (Section C) and present additional results (Section D), on IHDP (D.1), ACIC2016 (D.2) and Twins (D.3).

## A   Extended background on CATE estimation

We begin by giving an extended problem definition to supplement the brief description given in Section 2. Throughout, we assumed we observe a sample $\mathcal{D} = \{(Y_i, X_i, W_i)\}_{i=1}^n$, with $(Y_i, X_i, W_i) \overset{i.i.d.}{\sim} \mathbb{P}$. Here, $Y_i \in \mathcal{Y}$ is a continuous or binary outcome of interest, $X_i \in \mathcal{X} \subset \mathbb{R}^d$ a vector of possible confounders (i.e. pre-treatment covariates which could affect both treatment and outcome) and $W_i \in \{0, 1\}$ is a binary treatment, assigned according to a (usually unknown) propensity score $\pi(x) = \mathbb{P}(W = 1 | X = x)$. Using notation from the Neyman-Rubin potential outcomes (PO) framework [1], our main interest would lie in the individualized treatment effect $Y_i(1) - Y_i(0)$: the difference between the PO $Y_i(1)$ if treatment is administered ($W_i = 1$) and $Y_i(0)$ if individual $i$ is not treated ($W_i = 0$). However, only one of the POs is observed as $Y_i = W_i Y_i(1) + (1 - W_i)Y_i(0)$. Therefore, in line with most related work, we focus on estimating the conditional average treatment effect (CATE)

$$\tau(x) = \mathbb{E}[Y(1) - Y(0)|X = x] = \mu_1(x) - \mu_0(x) \qquad (1)$$

which is the expected treatment effect for an individual with covariate values $X = x$ and where $\mu_w(x) = \mathbb{E}[Y(w)|X = x]$ is the expected PO. The most prominent identification strategy for CATE in the ML literature is to assume that enough covariate information is measured to ensure that $\mathbb{E}[Y(w)|X = x] = \mathbb{E}[Y|W = w, X = x]$, which as formalized below, requires that treated and untreated units are exchangeable given treatment ("Unconfoundedness"), and that external assignment of a treatment leads to revelation of the associated PO ("Consistency"). Additionally, to enable nonparametric estimation of CATE, every individual needs to have non-zero probability to be assigned each treatment ("Overlap").

Together, these assumptions give the standard but *untestable* ignorability assumptions [2] in the PO framework:

**Assumption 1.**  *[Consistency, unconfoundedness and overlap]*
*Consistency: If individual $i$ is assigned treatment $w_i$, we observe the associated potential outcome $Y_i = Y_i(w_i)$.*
*Unconfoundedness: there are no unobserved confounders, so that $Y(0), Y(1) \perp\!\!\!\perp W | X$.*
*Overlap: treatment assignment is non-deterministic, i.e. $0 < \pi(x) < 1, \forall x \in \mathcal{X}$.*

35th Conference on Neural Information Processing Systems (NeurIPS 2021) Track on Datasets and Benchmarks.

## A.1 What makes CATE estimation special?

As in [3], we argue that there are three characteristics of the CATE estimation problem making it fundamentally different from standard supervised learning. In the main text, we focused on the third component, because – as we highlight below – this aspect persists even in an ideal randomized experiment.

**• 1. The neccessity to rely on untestable assumptions.** The fundamental difference in making *causal* claims relative to *associational* claims lies in the necessity to use *expert domain knowledge* to argue whether a treatment effect is identifiable from observational data [4]. Once *identifiability* has been established (e.g. by confirming that all possible confounders of a treatment-outcome relationship have been measured), all remaining problems are *statistical* in nature as they concern *estimation*.

**• 2. Confounding induces covariate shift.** Even when all confounders are observed, if the propensity score $\pi(x)$ is not constant, then the distribution of covariates in treatment and control group will differ, and not be equal to the marginal distribution of covariates across groups (the target distribution). When (1) is estimated by fitting separate functions for each $\mu_w(x)$ using empirical risk minimization, this can be problematic because the observed samples exhibit *covariate shift* [5] (asymptotically, this matters only if models are misspecified [5–7]). Such covariate shift problems are prevalent also outside the treatment effect literature, and have been studied in depth in the (unsupervised) domain adaptation literature (see e.g. [8–10]).

**• 3. The target label of interest is absent.** In fully randomized experiments, identifying assumptions hold by construction and the distribution of covariates across treatment arms is identical (in expectation) – yet CATE estimation remains non-trivial. This is because the true target label $Y(1) - Y(0)$ is absent even in experimental studies. An exception to this would be studies in which $X$ is discrete and low-dimensional; in this case, multiple observations can share the same covariate values $X = x$, which would allow to directly construct empirical estimates of CATE within each $X$-cell. In this paper, we focus on the more challenging case where $X$ contains continuous variables and/or is high-dimensional, so that no two observations can be expected to have identical $X$-values. Of course, because $Y(1)$ and $Y(0)$ are available separately to perform outcome regressions, $\tau(x)$ can be estimated indirectly as $\hat{\mu}_1(x) - \hat{\mu}_0(x)$. However, as we discuss in the main text, if $\tau(x)$ is a much simpler function than each $\mu_w(x)$, this approach will lead to much slower convergence than if the true target $Y(1) - Y(0)$ was available.

## A.2 Additional overview of relevant methods

Here, we give a more detailed and technical overview of the CATE estimation methods considered in the case studies. As in the main text and in [3, 11], we make a high-level distinction between *indirect* and *direct* estimation strategies.

The former (indirect estimators) refers to strategies that target the POs and estimate $\tau(x)$ as $\hat{\mu}_1(x) - \hat{\mu}_0(x)$. The S- and T-learner discussed in [12] are two model-agnostic ('meta-learner') learning strategies that estimate CATE *indirectly*. The S-learner fits a **s**ingle regression model $\hat{\mu}(x, w)$ by concatenating the covariate vector $X$ and the treatment indicator $W$ into $X'$ and then regressing $Y$ on $X'$, providing a final CATE estimate indirectly as $\hat{\tau}(x) = \hat{\mu}(x, 1) - \hat{\mu}(x, 0)$. The T-learner fits **t**wo regression models (one $\hat{\mu}_w(x) = \mathbb{E}[Y|W = w, X = x]$ for each treatment group) *separately* using only observations for which $W = w$, and provides a final CATE estimate as $\hat{\tau}(x) = \hat{\mu}_1(x) - \hat{\mu}_0(x)$. We consider [13]'s TARNet a hybrid between S- and T-learner, as some, but not all, information is shared between the two PO estimators: Formally, TARNet operates by jointly learning a shared feature map $\Phi : \mathcal{X} \rightarrow \mathcal{R}$, and two PO-specific regression heads $h_w : \mathcal{R} \rightarrow \mathcal{Y}$ (fit using only the data of the corresponding treatment group) each parametrized by a NN. The output heads are then used for indirect estimation of CATE as $\hat{\tau}(x) = h_1(\Phi(x)) - h_0(\Phi(x))$.

The latter (direct estimators) refers to strategies that output an estimate of $\tau(x)$ directly, *without* explicitly relying on the $\mu_w(x)$. In our experiments, we focused on the (model-agnostic) R-learner strategy [14] as an example for direct learners. The R-learner is based on [15]'s approach for semiparametric regression, and uses orthogonalization with respect to the nuisance functions $\pi(x)$ and $\mu(x) = \mathbb{E}[Y|X = x]$ (the unconditional outcome expectation). The algorithm proceeds in two stages; the first stage obtains estimates $\hat{\pi}(x)$ and $\hat{\mu}(x)$, which are then used in a second stage

estimating $\tau(x)$ directly based on the following loss:

$$\arg\min_\tau \sum_{i=1}^{n} \left[\{Y_i - \hat{\mu}(X_i)\} - \{W_i - \hat{\pi}(X_i)\}\tau(X_i)\right]^2 + \mathcal{R}(\tau(\cdot)) \tag{2}$$

where $\mathcal{R}$ is a regularizer for the complexity of $\tau(x)$. There exist multiple other two-stage direct estimation strategies (see [11, 12, 16]); here we focused on the R-learner because [17]'s causal forest solves a local moment equation based on the Robinson transformation which is analogous to (2).

## B  CATE benchmarking practice outside the ML literature

In this section, we contrast ML benchmarking practices for CATE estimation with practices that seem to be established in other literature (e.g. statistics and related fields). A core aspect of benchmarking practice in ML at large seems to be to conduct 'horse-races'[18] on pre-determined, passive and relatively opaque datasets to establish a proposed method as the new 'state-of-the-art'; often sources of gain are not sufficiently analysed [19]. The CATE estimation literature within ML inherits some of these practices, in particular, the frequent and somewhat passive reuse of benchmarks like IHDP and ACIC2016, on which only aggregate scores are reported without deeply engaging with key drivers of performance differences (as discussed in the main text). Nonetheless, not all papers rely solely on opaque benchmark comparisons; many recent papers report additional simulation studies to analyze the effect of some underlying problem dimension (e.g. [11, 20, 21]).

In other fields, empirical evaluation across a wide range of paper-specific simulation settings (covering many different, yet possibly stylized, conditions an estimator might encounter in the wild) seems much more common than ML's focus on replication of results on a specific benchmark. In statistics in medicine, for example, it seems to be common to conduct a range of simulation experiments probing different dimensions of the problem, and then to present some qualitative analysis of real data; both when new methods are proposed [22–24] and in benchmarking studies [25–27]. This was also how [28] originally used the IHDP simulations: the now popular benchmark (setting B: 'nonlinear and not parallel across treatment conditions') is only one of three simulation settings[1] considered in [28], after which [28] also considered a *real* application using original IQ scores from the IHDP experiment as outcomes as an illustrative example. Evaluation using a range of hand-crafted DGPs seems to be common also in other areas of statistics [12, 14, 17, 29].

Note that this section serves a purely descriptive purpose: we do not wish to imply that the benchmarking practices in other fields are necessarily superior to or even fully appropriate for the ML method development context. For example, inclusion of *qualitative analyses* of real data may be unnecessary for a methods paper and even inappropriate if insufficient domain knowledge is available to correctly interpret results (see [18] for an in-depth discussion of this in a different context). We do, however, think that ML benchmarking practice for CATE could benefit from more principled simulation studies as they are conducted in the statistics literature.

## C  Additional implementation details[2]

We use all random forests with standard hyperparameters as implemented in `grf`[3][30]; in particular this entails using a very large number of trees (2000). All neural networks are used with standard hyperparameters and components as implemented in `catenets`[4] for [11]; these are in turn based on the hyperparameters used in [13] for the IHDP experiments. In particular, all networks use dense layers with exponential linear units (ELU) as nonlinear activation functions, are trained with Adam, minibatches of size 100, and use early stopping based on a 30% validation split. All estimators have 3 representation layers of 200 units, 2 hypothesis layers with 100 units and a final prediction layer, and a small l2-penalty is applied to all weights. Refer to [11] for further detail. Finally, we use RNet without cross-fitting.

---

[1]There are also setting A:'linear and parallel across treatment groups', setting C: also 'nonlinear and not parallel' but without shared coefficients across $\mu_w(x)$, and variations of A & B with different error distributions.

[2]Code to replicate all experiments is available at `https://github.com/AliciaCurth/CATENets`

[3]Available on CRAN or at `https://github.com/grf-labs/grf`, under GPL-3.0 License

[4]Available on pypi or at `https://github.com/AliciaCurth/CATENets`, under BSD-3-Clause License

We retrieved the IHDP-100 data from `https://www.fredjo.com/`. We retrieved the ACIC2016 competition data from `https://jenniferhill7.wixsite.com/acic-2016/competition`; the transformations for Fig. 4 were performed using the script `https://github.com/vdorie/aciccomp/blob/master/2016/R/transformInput.R` from the competition R-package.

All experiments can be replicated with the code provided at `https://github.com/AliciaCurth/CATENets`. Experiments with grf were conducted using R 3.6 on Windows OS and experiments with catenets were conducted using Python 3.8 on Ubuntu 20.04 OS, with a Intel i7-8550U CPU with 4 cores, and running all IHDP experiments (2 times 100 runs with 5 replications for 6 considered algorithms) took between 12-24h.

# D Additional results

## D.1 Additional IHDP results

In Fig. 1 (RFs) and 2 (NNs) we present full results on the two IHDP settings, which we left out of the main text for readability. Performance differences appear more striking for RF-based than for NN-based methods.

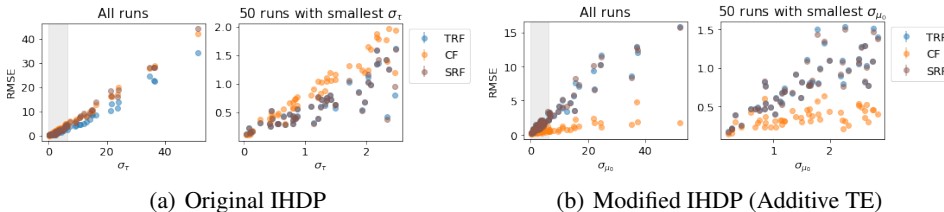

(a) Original IHDP        (b) Modified IHDP (Additive TE)

Figure 1: Out-of-sample RMSE of CATE estimation across 100 IHDP draws (original and modified setting) for forest-based estimators. Averaged across 5 runs, bar indicates one standard error. Shaded area in left plots indicates area which are zoomed on in right plots.

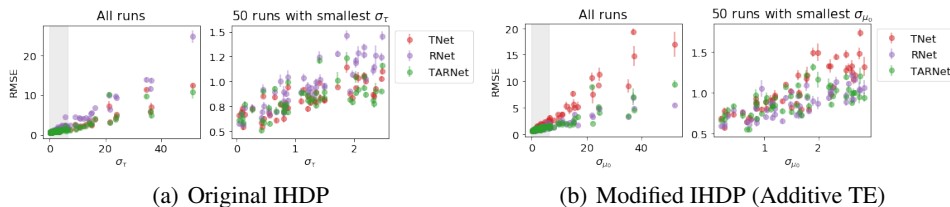

(a) Original IHDP        (b) Modified IHDP (Additive TE)

Figure 2: Out-of-sample RMSE of CATE estimation across 100 IHDP draws (original and modified setting) for NN-based estimators. Averaged across 5 runs, bar indicates one standard error. Shaded area in left plots indicates area which are zoomed on in right plots.

## D.2 Case study 2: ACIC2016

The datasets used in the Atlantic Causal Inference Competition (ACIC) 2016 are based on real covariates ($n = 4802, d = 58$) from the Collaborative Perinatal Project. The competition organizers created 77 simulation settings which varied in the functional form of the response surfaces, and the degrees of confounding, overlap and TE heterogeneity (see [31] for more detail); also here the main goal was to estimate the ATT. These datasets were used in e.g. [21, 32–34] to evaluate CATE estimators. Once more, our main interest lies in how the PO specification in the DGP influences the observed relative performance of algorithms. Therefore, we consider only a subset of settings and fix all 'experimental knobs' except for the degree of TE heterogeneity (which determines the similarity of the POs). Here, we focus on settings 2, 26 and 7, which have exponentials in their response surfaces and differ only in that they have no, low and high TE heterogeneity, respectively. We chose this triplet as it is the only one which has all three heterogeneity settings available. For each setting, we present out-of-sample results for the first 10 (out of 100) simulation runs provided by [31],

where we use the first 4000 observations for training and the remaining 802 for testing. Because of their higher variability, we average all NN results across 10 replications instead of 5.

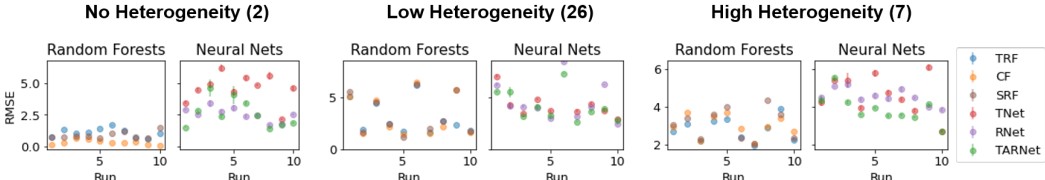

Figure 3: Out-of-sample RMSE of CATE estimation across 10 runs of a no, low, and high heterogeneity simulation setting. Averaged across 5 (10 runs for NNs) runs, bar indicates one standard error

### D.2.1 Empirical results

•**Finding (i): There is substantial variation in absolute performance across different runs of *the same* setting.** We again make a general observation on distribution of RMSE scores. Similar to the IHDP dataset, there is substantial variation in absolute performance of algorithms across different runs *of the same setting*. As can be seen in Fig. 3, when considering the performance of the forest-based estimators, the differences in *absolute* performance of *the same* algorithm across different runs are often larger than the differences between different strategies (using the same underlying ML method) on the same run. We conjecture that this is because [31] randomly sample terms which enter the response surfaces in each run, making some runs randomly harder than others. Similar to the IHDP dataset, simple averaging across runs can therefore mask differences that are visible mainly on the run-level.

•**Finding (ii): Relative performance of direct and indirect learners across heterogeneity settings varies as expected.** In Fig. 3, we observe that there is a trend across the three heterogeneity settings: In absence of heterogeneity, direct learners have a clear advantage and S-learners outperform T-learners. For low heterogeneity, all methods show similar performance. For high heterogeneity, the observations are reversed and indirect learners generally perform best. This difference is much more consistent and clear for the tree-based estimators than for the NN-based estimators. Further, TARNet, as a hybrid between S- and T-learner strategy, seems to inherit both their advantages, and can even match the performance of RNet on the setting without HTE.

•**Finding (iii): Underlying simulation favors tree-based methods.** Despite looking at a setting with exponential response surfaces and abundant data, we observe that in Fig. 3, RFs outperform the NNs – which stands in apparent contrast to the findings in Case study 1. Finally, we therefore investigated whether there is a reason for this, hidden in the DGP. We found that the simulated response surfaces are *not* created using the raw data as input, instead the 27 count variables are dichotomized by [31] before they are used in the DGP. This is the most natural pattern to represent for a RF, but we speculated that the NNs may struggle with this. To test this hypothesis, we feed the transformed data as used in the DGP by [31] to both RFs and NNs. In Fig. 4, we observe that indeed the performance of the RFs does not change, while the NNs improve substantially. This highlights that also here, one ML method is a priori better equipped to model the response surfaces than another.

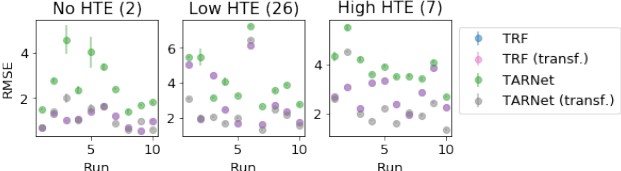

Figure 4: Out-of-sample RMSE of CATE estimation for TRF and TARNet with and without pre-transformed data. Purple dots are a consequence of blue and pink dots overlapping perfectly.

### D.2.2 Conclusion Case study 2

We found that, in the considered simulations of ACIC2016, the relative performance of different learning strategies across heterogeneity settings is also in line with expectations. As the 77 settings provided by [31] consist of only 2 settings without heterogeneity, 32 with low and 43 with high heterogeneity, we would expect that – on average – indirect learners will generally be favoured on

this benchmark. Further, we found that some aspects of the underlying DGP may inherently favor tree-based methods.

### D.3 Case study 3: Twins

In this section, we investigate the Twins benchmark of [35] – which is a rare example of a *real* 'all potential outcomes' dataset – in which twin pairs are assumed to represent counterfactuals. Here, the treatment is 'being heavier at birth' and the outcome is one-year mortality. We replicate the experimental setup of [3] and study the performance of different estimators as $n_{train}$ increases (while treatment is assigned fully randomly with $p_{treat} = 0.5$). Refer to Appendix E.2 of [3] for a more extensive overview of the experimental setup.

[3] note that because the outcome is binary and imbalanced (infant mortality is fortunately low), the signal for presence of treatment effects is relatively weak and noisy (as Twins with opposite outcomes are rarely observed due to the binary nature of the outcome). We therefore hold out $50\%$ of all data (5700 pairs of Twins) for evaluation and, as in [3], we use shallower networks (1 hidden layer each for representation and hypothesis) than for the experiments in the main text.

#### D.3.1 Empirical results

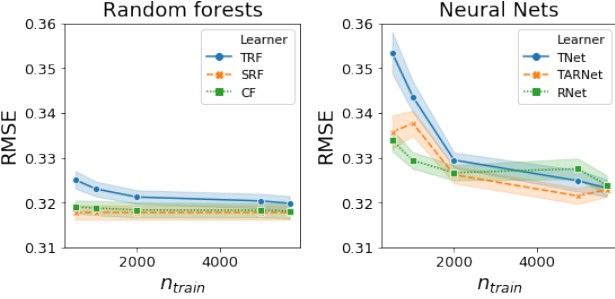

Figure 5: Out-of-sample RMSE of $\hat{\tau}(x)$ (relative to the counterfactual difference $Y(1) - Y(0)$) by $n_{train}$ on the Twins dataset. Averaged across 10 runs, shaded area indicates one standard error

•**Finding (i): Comparing ML methods, forests appear to perform best.** In Fig. 5 we evaluate $\hat{\tau}(x)$ against the true $Y(1) - Y(0)$ for different values of $n_{train}$. We observe that the considered RFs significantly outperform the NNs, particularly for $n_{train}$ small. We believe that the data-modality is likely related to this finding, as most covariates in the Twins dataset are *categorical*, which is more natural for RFs to model. Additionally, NNs are more sensitive to hyperparameters than RFs, so that considering different hyperparameter settings, e.g. further restricting the capacity of the NNs for smaller $n_{train}$, may lead to improved performance of NNs.

•**Finding (ii): Comparing estimation strategies, S- and R-learners appear to perform best.** Within both ML methods, we observe that T-learners perform significantly worse than R- and S-learner variants (recall that we consider TARNet a hybrid between S- and T-learner), while the latter two perform very similar. Although this may appear somewhat counterintuitively at first, as both S- and T-learner are indirect estimators, these findings are reminiscent of simulation experiments conducted in [12]: [12] demonstrate that when there is *no* treatment effect heterogeneity, S-learners perform very well, as they can effectively learn to *ignore* the treatment indicator (so that $\hat{\mu}(x, w) \approx \hat{\mu}(x)$). In Fig. 6 we plot the distribution of predicted $\hat{\tau}(x)$ across the test-set and observe that indeed, SRF predicts almost no CATE heterogeneity. The heterogeneity predicted by CF is somewhat larger, while TRF makes much more heterogeneous predictions. This observation, paired with the relative performance of the learners, provides some evidence that the treatment effect heterogeneity in the Twins dataset may be very low.

#### D.3.2 Conclusion Case study 3

In this section, we have studied the performance of the different algorithms on the *real-world* dataset Twins. Within the considered learning algorithms, RFs outperformed NNs, and S- and R-learners outperformed T-learners. The Twins benchmark thus appears to contain a completely different setting

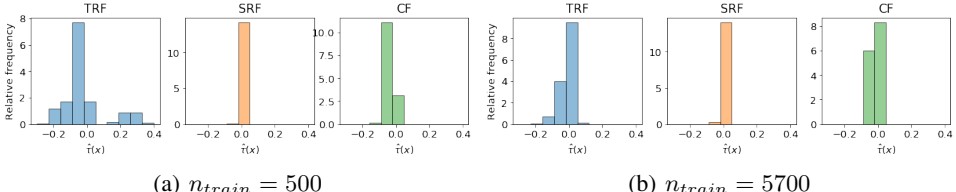

(a) $n_{train} = 500$          (b) $n_{train} = 5700$

Figure 6: Histogram of predicted CATE (test-set) for TRF, SRF and CF, for one run with $n_{train} = 500$ (left) and one run with $n_{train} = 5700$ (right).

than the majority of IHDP and ACIC2016 simulations, where treatment effect heterogeneity is very high and relative performances are often the reverse of what we found here. These findings, while very interesting, should nonetheless only be considered anecdotal evidence for the absence of major heterogeneity in real applications: as we argued in the main text, a major shortcoming of using real data relative to simulations is that each benchmark can only give insights into *one specific* real-world scenario (and even this is reliant on the strong assumption that twins are indeed a good counterfactual proxy), thus the observed superior performance of algorithms predicting low heterogeneity cannot necessarily be generalized to other applications. Yet, e.g. [12] and [36] do argue that CATE is often a simple function (relative to the POs) in practical applications; the findings presented here could provide some anecdotal support for this argument. By curating a broader range of such real-world 'all potential outcomes' datasets, future work could test whether this observation holds across different applications and datasets – which could in turn inform researchers creating new (semi-)synthetic datasets on the likely degree of CATE heterogeneity.