# OpenReview forum: "Really Doing Great at Estimating CATE? A Critical Look at ML Benchmarking Practices in Treatment Effect Estimation"
_NeurIPS.cc/2021/Track/Datasets_and_Benchmarks/Round2 — NeurIPS 2021 Datasets and Benchmarks Track (Round 2)_

### Official Review · Reviewer_ZoGp · 2021-09-01
**Review of the paper**

**Rating:** 7
**Confidence:** 4

**Strengths:**

I think the biggest strength of the paper is to provide in-depth analysis on the dominant benchmark semi-synthetic dataset, IHDP data (and ACIC data in the appendix). In particular, the authors argue that the DGP underlying the IHDP benchmark indeed favours both (i) NNs over RFs and (ii) indirect over direct estimators. There are also some other issues, such as the long and heavy tailed metric score and difference of relative performance across runs. The discussions surrounding these investigations are summarized in Section 3. Given that the current CATE estimation community heavily depends on IHDP, I think this analysis (Section 3) is worth publishing. The work will make the researchers in the area rethink how to evaluate/compare the empirical performance of CATE algorithms.

Other contributions include the alternative benchmark guidelines and evaluation metrics. Having re-evaluation based on these proposals could strengthen the contributions more.


**Weaknesses:**

The authors summarize alternative evaluation metrics for CATE estimation methods in Section 5, which is itself great. However, I think that most of the readers expect the additional benchmark results on conventional estimators based on the proposed metrics.

Some figures are very small (e.g., Figures 1 and 4) and I was not comfortable when trying to understand what they were saying. I suggest the authors improve the presentation of the simulation results leveraging the one additional page in the camera-ready version (if the paper gets accepted)

One thing I want to ask just out of my curiosity is about one of the alternative metrics, the decision quality (“Do estimators lead to correct treatment rules?”). I think the authors are aware of another related literature on batch bandit learning from logged bandit data, which aims at directly optimizing the decision quality from observational data (or logged bandit data). The batch bandit learning algorithms are evaluated based on their decision quality in nature. So, I would like to ask how the authors differentiate these two literature (CATE estimation and off-policy learning). Given that the authors think that one of the goals of the CATE estimation is to optimize decision making, then do you think the off-policy learning algorithms should be included as baselines when the decision quality metric is used? Or, do you see that these two approaches have completely different motivations?

**Additional Feedback:**

For suggestions for improvement, please see the “Weaknesses” section.

**Clarity:**

The contents of the paper are easy to follow and the motivation is convincing. Additional results are summarized in the supplementary materials in detail.

**Correctness:**

The analysis on the semi-synthetic data seems to be well-designed by the authors and the experiments are able to capture the flaws and concerns of the existing benchmark semi-synthetic datasets.

**Documentation:**

The simulation designs are well documented in the supplementary materials (Appendix C). The code used to produce the results (analysis of the previous datasets) are also included with a documentation, which is nice. I suggest the authors publicize the replication code on a GitHub repository so that other researchers can have easy access.

**Ethics:**

Does not apply

**Relation To Prior Work:**

The paper has an in-depth investigation of the current simulations study, which is one of the main contributions. The paper thus has sufficient explanation of the current literature and conventional way of benchmarking CATE estimation methods. The citations also look sound. Section 2 and Appendix A well summarizes the basic formulation and background of the literature.

As I noted above, it might be better to add some explanation about the connection with the off-policy learning literature in "Do estimators lead to correct treatment rules?" in Section 5. Would love to see the authors' thoughts on this.

**Summary And Contributions:**

The authors argue that CATE estimation algorithms have been evaluated only on a very limited set of semi-synthetic benchmark datasets, even though the literature grows rapidly. So the authors investigate current benchmarking practices for ML-based CATE estimators, with special focus on empirical evaluation based on the popular semi-synthetic IHDP benchmark. The case studies found problems with current practice and highlight that inherent characteristics of semi-synthetic benchmark datasets can favor some algorithms over others. There are also some other issues that should be taken into account in empirical studies. Finally, the authors discuss alternatives for current benchmark datasets, and implications of our findings for benchmarking in CATE estimation.

---

> ### Author Response · Authors · 2021-09-24
> **Response to Reviewer ZoGp**
>
> Dear Reviewer ZoGp, thank you for your thoughtful and in-depth review! We are delighted by your appreciation of our work! Below, we respond to points raised in the review.
>
> (A)	**Evaluating alternative metrics:** We decided to not evaluate the discussed metrics on current benchmarks, because – as we discuss in l. 377f – systematically evaluating such metrics requires benchmarks with appropriate `experimental knobs’. Further, neither IHDP nor ACIC2016 can be directly used to evaluate the discovery of drivers of effect heterogeneity, as variable status is not recorded in the benchmarks. Even if such variable status was available, the DGP used in IHDP would not be particularly interesting for this task, as all important variables are simultaneously predictive and prognostic.
>
> (B)	**Figure size:** We will most certainly ensure that Figures 1 and 4 receive more space once we can use the additional content page in the final manuscript. Thank you for pointing this out!
>
> (C)	**Comparing with off-policy learning algorithms.** We would argue that it in general should not be necessary to compare the quality of decision rules implied by CATE estimators to baselines from the off-policy learning literature (which usually do not provide estimates of CATE itself), as (in a CATE estimation paper) this additional metric should only serve the purpose of evaluating the quality of CATE estimators along an _additional_, practically relevant, dimension (providing decision support/treatment recommendations). That being said, we do think that a benchmarking study comparing treatment recommendations made by state-of-the-art CATE estimators to off-policy learning algorithms might be interesting in its own right. Thank you for this interesting question!

---

> ### Author Response · Authors · 2021-09-27
> **The paper has been updated to reflect the reviews**
>
> Once more, we would like to thank you for your invaluable feedback. We have just uploaded a revision of our paper, in which we -- as per your suggestion -- used the additional space to increase the size of Figures 1 and 4; we hope that this makes the paper easier to read! Additionally, we were wondering whether our responses (Sep 24) have addressed your concerns? Should you have any leftover comments or concerns, please do let us know - we would be happy to do our utmost to address them in the limited time left in this response period!

---

> > ### Comment · Reviewer_ZoGp · 2021-09-28
> > **Read your comments**
> >
> > Thank you for your reply and the paper update. Better than the last version, I think. Nice to have your answers to my questions.

---

### Official Review · Reviewer_X29q · 2021-09-17
**Useful results but with limited contribution.**

**Rating:** 4
**Confidence:** 4
**Correctness:** Claims apears to be correct in this p…
**Clarity:** The paper is well-written and easy to…

**Strengths:**

(1)	Experiments are conducted on IHDP, and several useful conclusions are presented.
(2)	The paper emphasizes synthetic datasets have problems in providing fair and insightful benchmarking results. This is a key problem of existing research.


**Weaknesses:**

I think this paper provides limited contribution as a benchmarking paper:
(1)	Experiments are only conducted on one dataset. This seems not thorough enough for a benchmarking.
(2)	This paper presents the disadvantage of using synthetic datasets. But no solution or new datasets are provided. This limit the contribution the this paper, and this key conclusion is somehow straightforward.


**Additional Feedback:**

How to construct a better non-synthetic dataset for the CATE problem?

**Documentation:**

There is sufficient detail to support reproducibility.

**Ethics:**

There is no ethical concerns.

**Relation To Prior Work:**

It is clearly discussed how this work differs from previous contributions.

**Summary And Contributions:**

This paper presents a benchmarking study for estimating conditional average treatment effect. Experiments are conducted on IHDP, and several useful conclusions are presented. The paper emphasizes synthetic datasets have problems in providing fair and insightful benchmarking results.

---

> ### Author Response · Authors · 2021-09-24
> **Response to Reviewer X29q**
>
> Dear Reviewer X29q, thank you for your thoughtful review. We are grateful for your appreciation of our experiments, conclusions and writing! Below, we provide point-by-point responses to statements made in the review.
>
> (A)	_”This paper presents a benchmarking study”_ and  _”I think this paper provides limited contribution as a benchmarking paper”_. We want to emphasize, as we have written in the paper, that this paper is _not_ a benchmarking study. Rather, as we write in the abstract “we investigate current benchmarking practices” and “we identify problems with current practice”. This paper falls perfectly within the scope of the NeurIPS benchmarking track: not as a benchmarking paper, but rather as a paper ”identifying significant problems with existing datasets and their use” (this is a direct quote from the Call for Papers of this track – see section on scope in https://neurips.cc/Conferences/2021/CallForDatasetsBenchmarks).
>
> (B)	_“Experiments are only conducted on one dataset, which seems not thorough enough for benchmarking”_ . We reiterate that we are specifically investigating problems in current benchmarking practice using the IHDP benchmark and are not conducting a benchmarking study. It is _prescribed_ by the nature of our paper that we put our focus on one central dataset. However, to reinforce some of our conclusions, we *do* present additional results on another dataset  (the ACIC2016 benchmark) in appendix D2 (see reference to these results on l.59 and l.266f).
>
> (C)	_”But no solution or new datasets are provided.”_ The main purpose of this paper is to provide a critique of current benchmarking practices using the IHDP dataset; it does not intend to provide a new solution or new dataset. We _do_ provide (in Section 4) a concise overview of alternative benchmarking solutions as well as an impartial analysis of their inherent (dis)advantages. We hope that this can serve as a starting point for researchers looking to *use or develop new benchmarks*. We do not make a final judgement as to which of the discussed alternatives is best because we think that all alternatives for benchmarking datasets come with inherent limitations. As we argue in our conclusion “No type of dataset is without flaws, thus acknowledging limitations is essential“ and “Conducting experiments across multiple types of benchmarks .. may lead to the most convincing conclusions”(l.414f).

---

> ### Author Response · Authors · 2021-09-28
> **Did our responses address your concerns?**
>
> Once more, we would like to thank you for your feedback! We were wondering whether our responses (Sep 24) have addressed your concerns? Should you have any leftover comments or concerns, please do let us know - we would be happy to do our utmost to address them in the very limited time left in this response period!

---

### Official Review · Reviewer_9ShC · 2021-09-21
**Excellent, useful, critical, and constructive paper on current practices in evaluating CATE estimators**

**Rating:** 9
**Confidence:** 4
**Correctness:** They appear to be correct.

**Strengths:**

As noted above, the paper has many strengths. I appreciated the lengthy discussion of alternatives to synthetic benchmark data (Section 4), in which the authors give a clear discussion of strengths and weaknesses to proposed alternative benchmark dataset generating approaches, allowing the researcher a detailed guide to find an approach most suited for their setting/question.

I also appreciated the attention paid to alternative evaluation metrics (Section 5). Better metrics would help bring the evaluation of CATE estimators potentially more in line with the purpose of those estimators, for example in identifying individual characteristics with prognostic versus predictive value, and  looking at how well the estimator performs at making real-world predictions.

I appreciate the guide the authors provide for how a researcher might continue to use existing benchmarks with (semi-)synthetic data in a more thoughtful way, for more limited, carefully-chosen applications, and with an awareness of and discussion about the limitations.

Finally, I appreciated the suggestions for how benchmarking and dataset-generating practices could be improved by collaboration with domain experts.

**Weaknesses:**

Authors should be careful to define acronyms before using.

Given its importance to the paper, should define the acronym IHDP (Infant Health and Development Program) when it is first used, rather than on page 4 after the acronym has been used multiple times.

Similarly, "DGP" is used before it is defined, and it is defined incorrectly: Line 114: "The most common approach for simulating outcomes seems to be to rely on data generating processes (*DPGs*) with additive effects"

**Additional Feedback:**

None

**Clarity:**

Extremely well written; I found it pleasurable to read. The one major caveat is the issue with jargon-y acronyms like DGP being used before being defined. Because I'm a researcher with work in this space, this did not affect readability for me much, but I imagine it could affect the paper's readability for some others.

**Documentation:**

Yes

**Relation To Prior Work:**

Prior work is discussed. The authors could perhaps be clearer on which of their results, for example on data generating processes being highly influential to what kind of algorithm performs well on a popular benchmark, have been documented elsewhere in any form. For example, I was curious about how much overlap [23] has with their work.

**Summary And Contributions:**

"Really Doing Great at Estimating CATE? A Critical Look at ML Benchmarking Practices in Treatment Effect Estimation" is an eminently practical and extremely readable paper documenting and theoretically grounding many issues with benchmarking practices for ML-based CATE estimators. It is critical of current practices, but also constructive.

I found the paper to be extremely readable and am sure I will refer back to it in part as a literature review and guide to CATE estimation across fields. I particularly appreciated the synthesis and perspective the authors provided across disciplines, allowing the reader to gain a more unified understanding of CATE estimation across econometrics, statistics, and ML as a part of reading the paper.

The central findings of the paper, that popular benchmark datasets will favor certain algorithms over others in a way that is problematic for the field if it continues to blindly use these benchmarks, are clearly presented and persuasive. The authors provide the reader with enough theory to grasp exactly where and how those issues arise from the data generating process, and thus provide a framework for researchers who still wish to use a popular benchmark to properly contextualize their findings, tweak certain experimental knobs to check robustness, etc.

---

> ### Author Response · Authors · 2021-09-24
> **Response to Reviewer 9ShC**
>
> Dear Reviewer 9ShC, thank you for your thoughtful and in-depth review! We are delighted by your positive perception and appreciation of our paper!
>
> To address the points raised in the review, we will most certainly ensure that all acronyms are defined upon first use in the updated version. Further, in terms of related work, we consider the overlap with [23] minimal, as said paper focusses on _real world_ benchmarks for natural language processing, computer vision, reinforcement learning and transformers and hence does not consider (semi-)synthetic DGPs as used in the CATE literature. That being said, the authors do “show that relative model performance is highly sensitive to the choice of tasks and datasets it is measured on” (p. 3f) and postulate the existence of “task selection bias” in these fields; yet – given that these datasets are real -  we consider this a different type of bias than the problems we identify, which are induced by blindly relying on (semi)synthetic DGPs _as if they were real_.

---

> > ### Comment · Reviewer_9ShC · 2021-09-24
> > **Thank you**
> >
> > Thank you for the reply! I think it would be productive to include this clarification and distinction between your paper and theirs (and any others that are close to yours) in your literature review section.

---

> > > ### Author Response · Authors · 2021-09-27
> > > **The paper has been updated to reflect the reviews**
> > >
> > > Thank you for this suggestion, we have included it in the updated the related work section (l.86f)!

---

### Official Review · Reviewer_cjbh · 2021-09-21
**Some useful empirical results, but many conclusions are stronger than warranted**

**Rating:** 5
**Confidence:** 4

**Strengths:**

The paper is well written and well organized, and it is clearly motivated. The empirical results are compelling in that they directly address the motivating example, and they clarify and extend the previous results that are cited there. I think researchers working in this area, including those who are tasked with crafting future causal inference estimation challenges (which is one of the settings the authors cite), will be interested in these results. I think that Section 3 almost could stand alone as a research note in an appropriate publication venue.

**Weaknesses:**

While the empirical results are compelling, they point to a conclusion that seems to me fairly self-evident, which is that any given data generating setting will favor some estimators over others.

Finding (i), that the histogram of RMSE values over simulation runs has a long right tail, does not seem surprising, and plenty of papers CATE papers (e.g. Hill, 2011, which first proposed IHDP) also display the distribution of performance values over simulation runs rather than merely reporting averages.

Finding (ii) seems to me to be overstated. We can't conclude that "indirect" (aka plugin) estimators are better than "direct" (e.g. two-stage/doubly robust) estimators on IHDP, and we can't conclude that neural nets are better than random forests in this setting. we can only conclude that the particular learners the authors used (random forests and neural nets, with some set of hyperparameters) performed worse in the particular two-stage estimators they examined. In particular, the performance of the indirect estimators decomposes entirely into the performance of the constituent regressions, a point that the authors make in Section 2. So, when TNet outperforms TRF, for example, it presumably just means that a neural net did better at estimating one or both of the regression functions. Changing the hyperparameter settings could potentially reverse this relationship, so the statements about neural nets vs random forests don't seem warranted.

Findings (iii)-(iv) are interesting and useful, if not necessarily surprising. While they shed light on the previous findings wrt neural nets and random forests on IHDP that are cited as motivation, I don't think they yield any insight into any unique limitations of IHDP, or of (semi-)synthetic datasets in general. They simply show that different algorithms work well on different datasets, which is true of any real or synthetic data. The authors make this point themselves in the paper, but the abstract seems to promise a critique that is unique to semi-synthetic data.

The remainder of the paper is largely a summary of existing proposals for ways to construct benchmarks and to evaluate CATE estimators. Re: Section 5, the optimal treatment regimes literature typically uses measures other than RMSE to evaluate estimators, and, though I'm less familiar with this literature, I believe many papers also evaluate performance with respect to predictive or prognostic value specifically, when those are the target quantities. The use of RMSE as a performance metric naturally isn't appropriate to all scenarios, but it is certainly of interest in some scenarios. I'm not quite convinced that the literature has failed to distinguish these cases.

**Additional Feedback:**

No additional feedback

**Clarity:**

The paper is very clearly written and easy to read. My only comment here is that acronyms like DGP and PEHE should be spelled out when they are first introduced.

**Correctness:**

I believe the claims in the paper are all correct, with one exception.

Namely, I don't follow the logic behind the claim that the as the two regression functions become less additively similar, the direct estimators will increasingly outperform the indirect estimators. The CATE is at least as smooth as the less smooth of the two component regression functions, which is why direct estimators are advantageous in many settings (in an asymptotic sense). Rather, it may be that the asymptotic advantages of the direct estimators over indirect estimators diminish as the difference in smoothness between the CATE and the outcome regressions decreases. Things may of course behave differently in a finite-sample sense, but the claim that "...this is expected given the DGP..." under Finding (ii) does not seem correct.

**Documentation:**

N/A

**Relation To Prior Work:**

Yes, the work is clearly contextualized in the literature.

**Summary And Contributions:**

This paper examines a semi-synthetic data generating process, referred to as IHDP, that is frequently used to benchmark estimators of the Conditional Average Treatment Effect (CATE), which is a common target in causal inference. The authors compare several existing estimators/algorithms and show that their relative performance depends on the parameters of the data generating process, which vary randomly over simulation runs. They argue that IHDP naturally favors some estimation approaches over others, and that over-reliance on this benchmark may give the false impression that some estimators are uniformly better than others. They summarize other proposed approaches to benchmarking CATE estimators and discuss their advantages and disadvantages.

The primary contribution of this paper, in my view, is the empirical results, which yield some insight into how different algorithms perform on IHDP (although plenty of empirical results using IHDP exist already). However, the conclusions drawn with respect to IHDP in particular seem too strong, while the broader conclusions--that any benchmarking approach has limitations--seem fairly self-evident. Overall, I do not think the contributions are substantial enough to stand as a full paper at NeurIPS. [EDIT: following the author's remarks and revisions, I'm raising my rating from a 3 to a 5.]

---

> ### Author Response · Authors · 2021-09-24
> **Response to Reviewer cjbh (3/3)**
>
> (F)	**Overview of existing proposals.** We would like to argue that Section 4 is more than a mere summary of existing proposals for benchmarking CATE estimators. Instead, we aimed to give a _concise_ and _impartial_ analysis of the (dis)advantages of alternative benchmarking approaches, which we hope can serve as a starting and reference point for researchers looking to _use or develop new benchmark datasets_. We consider such impartial presentation an important complement to raising awareness of the shortcomings of current benchmarking practice (Section 3).
>
> (G)	**Metrics and OTR literature.** In Section 5, we specifically discuss metrics used in the _CATE estimation literature_, and hence did not intend to imply that the OTR literature uses RMSE metrics. Instead, the goal of this section was to highlight that the CATE estimation literature could _borrow_ metrics from e.g. the OTR literature to evaluate CATE estimators along additional dimensions, especially because personalized policy design/providing personalized treatment recommendations is often given as a motivation for estimating heterogeneous treatment effects in the first place [11].
>
> (H)	**Possible Incorrect claim.** We are in full agreement with the reviewer’s statement that “it may be that the asymptotic advantages of the direct estimators over indirect estimators diminish as the difference in smoothness between the CATE and the outcome regressions decreases”. This statement seems to be perfectly in line with our assertion in l.140f that the advantage of direct learners “diminishes as $f_1(x)$ and $f_0(x)$ become less similar”. We do agree that the highlighted passage in finding (ii) is imprecise in this regard, would be happy to rephrase l.223f as “This is not unexpected given the DGP and the theoretical arguments in the previous section: $\mu_0(x)$ and $\mu_1(x)$ are not similar (on an additive scale) in the underlying DGP, making $\tau(x)$ a difficult function to estimate directly. Thus, direct learners have no expected theoretical advantage in this setting, while the two-stage nature of such algorithms may hurt finite sample performance relative to indirect estimators as errors could accumulate across stages.“

---

> ### Author Response · Authors · 2021-09-24
> **Response to Reviewer cjbh (2/3)**
>
> (C)	**Histogram of RMSE values.** We would politely like to disagree with the statement that "plenty of CATE papers (e.g. Hill, 2011, which first proposed IHDP) also display the distribution of performance values over simulation runs rather than merely reporting averages”. First, note that while Hill (2011) does present the distribution of two performance metrics *regarding estimation of the ATT* (the bias in ATT estimates and interval length over runs) the paper reports only *summary statistics* for CATE estimation, i.e. the *average PEHE* across 1000 simulation replications, without any mention of the distribution of PEHE scores or its heavy right tail (see pages 14-15 in Hill, 2011). Second, we are not aware of ‘plenty of CATE papers’ that display more than mere averages (and their standard errors) across IHDP simulation runs; e.g. [1, 2, 3, 4, 5, 6, 7, 8, 9, 54] all report mean (SE) without mention of heavy tailed RMSE scores. The only ML paper that we are aware of that explicitly presents and discusses a histogram of RMSE scores across IHDP is reference [10] (Appendix, page 7), which is our own paper; the distribution of RMSE scores, which we had to defer to the Appendix as it was merely an ancilliary observation, was at the time of writing extremely surprising to us given the lack of discussion elsewhere in the literature and actually motivated the extended treatment we present here.
>
> (D)	**Statement of finding (ii).** The key motivation of this section was to understand the performance differences between TARNet and Causal Forests as they are presented in many CATE estimation papers as baselines; thus we relied on hyperparameter settings similar to those used to create the original results. We did not intend to imply that we can draw conclusions beyond the *considered estimators* (encompassing the estimation strategy, used ML methods and the used hyperparameters) in this paragraph, and will update the wording in the manuscript to reflect this limitation more clearly. That being said, regarding your specific example, we *do* indeed further investigate the sources of relative performance of RFs and NNs in finding (iv) and highlight that, due to the exponential specification in the _DGP used in IHDP_, RFs (which can be seen as adaptive nearest neighbor estimators) seem to have a boundary-bias/extrapolation issue - which would persist regardless of the used hyperparameters.
>
> (E)	**Uniqueness of critique to semi-synthetic data.** The critique that we present in this paper is aimed at *how* semi-synthetic data is *currently used* in CATE benchmarking practice, not at semi-synthetic data itself. As we tried to highlight in point (B) above and in the paper in e.g. l.52f (introduction), l.205f (section 3), l. 274f (section 4) or in l. 408f (conclusion), the problem we would like to highlight is that experiments on semi-synthetic benchmarks such as IHDP are currently performed *as if the data was real* (not simulated), without discussing whether their characteristics (in particular those that favour some algorithms over others) are likely to appear in practice. We thank you for highlighting that this point could be made more clear also in the abstract, and will update it in the final version.

---

> ### Author Response · Authors · 2021-09-24
> **Response to Reviewer cjbh (1/3)**
>
> Dear Reviewer cjbh, thank you for your thoughtful and in-depth review! We are very grateful for your appreciation of our writing, motivation and our empirical results! Below, we first discuss why we consider the NeurIPS benchmarking track an appropriate publication venue and then present point-by-point responses to your criticisms. We would like to preface our responses by highlighting that it was our intent to  position this piece as a _cautionary tale_ investigating the shortcomings of _the current use_ of semi-synthetic datasets in the benchmarking process _as if they were real_ and as such, only aimed to draw conclusions about possible problems in current benchmarking practice. As outlined below, we would be happy to rephrase certain passages to reflect this intent more carefully.
>
> (A)	**Appropriateness of publication venue.** We are delighted by your assessment that researchers working in this area “will be interested in these results” and “that Section 3 almost could stand alone as a research note in an appropriate publication venue.” This is indeed precisely why we wrote this paper; therefore, we would like to convince you that the NeurIPS benchmarking track *is* such an appropriate venue! In the call for papers *“identifying significant problems with existing datasets and their use”* is specifically listed as a topic within scope of this track (see section scope in https://neurips.cc/Conferences/2021/CallForDatasetsBenchmarks). The largest part of our paper, namely the empirical results presented in Section 3 and Appendix D, falls precisely under this description.  As the IHDP dataset has *mainly* been used in publications presented at ML conferences, e.g. NeurIPS [3, 26, 42], ICML [1, 2, 4, 25] and ICLR [5, 8], we would additionally argue that the NeurIPS benchmarking track targets exactly the right audience for this critique.  We decided to add Sections 4 & 5, which present the up-and downsides of alternatives for benchmarking, in addition to the critique of existing datasets in order to provide some guidance what researchers could do instead of relying on the (flawed) IHDP benchmark only.
>
> (B)	**Self-evident conclusion.** We agree with you that the conclusion that any DGP may inherently favor some algorithm may *seem* somewhat self-evident (Indeed, we state in our conclusion (l406f): “We do not consider this finding surprising, yet it is rarely taken into account (or is at least not explicitly discussed) when benchmark datasets and baseline algorithms are selected for use in related work.”). However, we consider the apparent ignorance for this phenomenon in experiments of virtually all related work in the ML literature relying on the IHDP benchmark as evidence that this problem is *not* “so obvious” to the ML community; therefore, we think that this point warrants explicit discussion to draw attention to the need to interpret empirical results more carefully. Specifically, in the context of the IHDP benchmark, we consider this conclusion important because the ML CATE literature appears to have been heavily relying on the benchmark without questioning (i) whether any algorithms are inherently (dis)advantaged by the simulated DGP and (ii) crucially, whether said (dis-)advantage is due to a feature of the DGP that is likely to appear in reality.
>
> We would argue that point (ii) is what makes the current benchmarking practice problematic: Hill (2011)’s setup B may inherently favour one set of algorithms, while the modified additive setup we consider in Figure 3 favours another – but which setup is more similar to what an algorithm would encounter in the wild?  The current practice of evaluating algorithms only on Hill (2011)’s setup B implicitly makes the judgment that this is a realistic or important setting (which, in turn, will influence which algorithms are regarded ‘state of the art’ and warrant further research) – however, as we describe in l.202f, we could not find any discussion within related work (neither in Hill (2011), who actually also reported performance on multiple other setups, nor in e.g. [1,2]) why this specific DGP is realistic within the considered application. On the contrary, with the exception of this DGP, the most common approach for simulating outcomes across different literatures (e.g. in statistics [15, 16, 35, 49], biostatistics[14, 20, 50] and econometrics[39]) seems to be to rely on *additive treatment effects* (cf l. 114f). We will try to make this point more clear in the updated manuscript.

---

> ### Author Response · Authors · 2021-09-27
> **The paper has been updated to reflect the reviews**
>
> Once more, we would like to thank you for your invaluable feedback. We have just uploaded a revision to reflect the changes outlined in our responses from Sep 24; we present a brief overview of relevant changes below. Additionally, we were wondering whether our responses combined with today's revisions have addressed your concerns? Should you have any leftover comments or concerns, please do let us know - we would be happy to do our utmost to address them in the limited time left in this response period!
>
>
> Summary of relevant changes:
> - We have updated the abstract (l.6f) and the introduction (l.53f) to emphasize that our critique is unique to the *current use* of semi-synthetic data because the data does not necessarily reflect characteristics of real data (but is treated during benchmarking *as if it was real*).
> - As requested by reviewer 9hSC below, we added a discussion of [23]'s finding that the relative performance of e.g. CV and NLP algorithms can also be sensitive to the choice of (real-word) dataset -- and how this differs from our setting -- in the related work section (l.86f).
> - We changed the statement of finding (ii) and the wording throughout section 3.1 to more carefully reflect that our conclusions apply only to the *considered estimators*
> - We updated the statement you identified as incorrectly phrased as described in point (H) in our response below.

---

> ### Author Response · Authors · 2021-09-29
> **Did our responses and updates address your concerns?**
>
> Dear Reviewer cjbh,
>
> As the discussion period ends today, we wanted to make a final enquiry whether our response (Sep 24) and paper update (Sep 27) have addressed your concerns. We would also be eager to use the very limited time left to follow up on any additional comments you may have!

---

> ### Comment · Reviewer_cjbh · 2021-09-29
> **Response to authors**
>
> Dear authors,
>
> Thank you for your thoughtful replies and revisions. I think the revised paper makes the scope and context of your critique much more clear; I appreciate clearly now that you are interested in the use of semi-synthetic datasets specifically in ML papers that exclusively deal with the CATE, and that your results are not necessarily intended to speak to benchmarking practices in closely adjacent literatures.
>
> I also appreciate the clarification about which parts of your critique are specific to semi-synthetic data. I find this point particularly helpful: "This is similar to our point that the DGPs in semi-synthetic benchmarks may inherently favour some algorithms, yet the key difference in our context is that such (dis-)advantages are systematic (partially foreseeable) and artificially created through the choice of DGP, instead of naturally inherent to a real application as in [23]."
>
> By "appropriate publication venue," I didn't mean that your paper seems unsuited for this track, just that I thought Section 3 could be presented as a more concise research note on its own somewhere. (It would probably be too short then for NeurIPS, is all I meant.) To be clear, your work seems like exactly the kind of work this track is intended for. My main hesitation was about whether the results are substantial enough for a full NeurIPS paper.
>
> I still have some hesitation about this, since I think that your empirical results speak fairly narrowly to the motivating example that compares causal forests to TARNet. However, while we agree that the fact that different DGPs naturally favor different algorithms seems self-evident, you have persuaded me that it is a point worth making explicitly, and your results certainly make that point.
>
> I think it would be very helpful to eventually expand your results to support the broader claims that you make in the paper. For example, since you point out that semi-synthetic datasets are not necessarily realistic, it would be very interesting and informative to repeat these experiments on the Twins dataset. With a broader set of results, I would have no hesitation about strongly recommending this paper.
>
> Just a couple remaining points that I think can be easily addressed:
> - Re: finite sample performance, I don't think it's true that direct estimators generally perform worse than indirect estimators on finite samples. My point was just that the advantages of direct estimators are asymptotic, so they may or may not manifest at any given sample size.
> - "We conclude that the IHDP benchmark has multiple flaws..." I'd describe these characteristics as limitations rather than flaws, unless you really think that IHDP is fundamentally unsuited to benchmarking. That doesn't seem like the point you make in the rest of the paper though.
> - A typo: "...we presented some evidence that would to speak": missing "seem"
> - Very minor: should probably capitalize Atlantic Causal Inference Conference
>
> Just to reiterate, I found your paper really well written and enjoyable to read, and the revisions certainly improved both my understanding and my overall rating wrt this venue.

---

> > ### Author Response · Authors · 2021-09-30
> > **The submission now includes Twins Results in the Appendix!**
> >
> > Thank you very much for engaging with us! We are delighted to hear that our clarifications and revisions addressed your major concerns! We are also grateful for your suggestion to repeat our experiments on the Twins dataset, and agree that this could constitute an informative supplement to the results we have already provided. As we discuss below, we were able to produce some interesting results on the Twins dataset today (see Appendix D.3 in the updated supplement), which we will expand on in the final version. Additionally, we also address your remaining points below and updated the main paper accordingly.
> >
> > We hope that these additional results and changes eliminate any residual hesitation on your part – if so, we would be very grateful if you would consider revising your score once more!
> >
> > (A)	**Results on Twins dataset**.
> > Constrained by the very limited time we had left in this response period, we have included results on the Twins dataset as well as preliminary discussions thereof in Appendix D.3 of the updated supplementary materials (replication code can also be found in the supplement). We would be happy to provide a more elaborate discussion of these results as well as more cross-references to them from the main manuscript in a possible camera-ready version. The results are indeed very interesting: as we discuss in more detail in Appendix D.3 we have found that, within the considered estimators,  RFs perform better than NNs, and S-learners&R-learners outperform T-learners.  The latter seems to be driven by the presence of very little evidence for treatment effect heterogeneity in the dataset; yet, as this dataset also represents *only one* specific possible real-world scenario (as discussed also in Section 4.2), this finding should be interpreted with caution.
> >
> > (B)	**Remaining points**. Thank you for highlighting these remaining minor issues; we have addressed them in the updated manuscript:
> > -	We did not intend to imply that direct estimators *generally* perform worse in finite samples in the updated passage; instead we meant that we could expect worse finite sample performance in the setting where $f_\tau$ is of similar complexity as $f_{\mu_w}$ (where both strategies would have comparable asymptotics). Thank you for highlighting that this was unclear; we have updated the phrasing in the manuscript to reflect this!
> > -	We changed “flaws” to “limitations”
> > -	We fixed both typos

---

> > > ### Comment · Reviewer_cjbh · 2021-10-04
> > > **Second response to authors; increased rating**
> > >
> > > [Dear Chairs: I am unable to edit my original review above to change the rating. Please consider my final rating a 6 rather than a 5.]
> > >
> > > [To the authors:] Thank you for your response and the additional analyses. These results are interesting and I think make the paper stronger. Since I think the scope of the contribution is still somewhat narrow (though broader than before), I'm increasing my rating to a 6.

---

### Decision · Program_Chairs · 2021-10-10

**Decision:**

Accept

**Comment:**

The final scores for this paper are 9, 7, 6, 4. There were some concerns about whether this paper was sufficiently significant given that it analyzes only one dataset, but the authors released results on Twins, and now the majority of reviewers recommends acceptance and there is a champion. The reviewer who recommended 4 also has not responded and has not commented on the new results on Twins. I have to say I disagree with the authors about their claim that the negative reviewer has issues with the fit of the paper for this track (I think their main criticism is about using only one dataset). In any case, I feel like there is enough support for this paper.